# Safety Monitoring Method for Pipeline Crossing the Mining Area Based on Vibration–Strain Fusion Analysis

**DOI:** 10.3390/mi16091074

**Published:** 2025-09-22

**Authors:** Jianping He, Tongchun Qin, Zhe Zhang, Ronggui Liu, Yuping Bao

**Affiliations:** School of Civil Engineering, Nantong Institute of Technology, Nantong 226000, Chinabaoyp_nt@126.com (Y.B.)

**Keywords:** pipeline safety monitoring, mining area, fiber Bragg grating sensor, distributed optical fiber vibration sensor, distributed optical fiber strain sensor

## Abstract

The overlying rock layers in a mining area may collapse or settle, subjecting pipelines to uneven forces that can lead to deformation or even fracture. This paper proposes a pipeline safety monitoring method that combines fiberoptic vibration and strain sensing to detect vibrations and deformations caused by rock layer collapse in mining zones. First, pipeline deformation monitoring under unknown force directions was investigated using fiber Bragg grating (FBG) sensing technology. Second, we constructed a mining area pipeline model and conducted vibration/deformation monitoring tests employing FBG sensors, distributed Brillouin strain sensing, and distributed fiberoptic vibration sensing technologies. The experimental results demonstrate that FBG sensor arrays deployed at 90-degree intervals can effectively identify the pipeline’s primary force direction and maximum strain, with direction angle errors of less than 5.2%. The integrated analysis of vibration and strain data enables accurate identification and measurement of extended vibration responses and pipeline deformations in open-air zones. This study establishes a comprehensive monitoring framework for ensuring pipeline safety in mining areas.

## 1. Introduction

During the construction and operation of long-distance oil and gas pipelines, buried pipelines offer advantages such as space savings, safety and stability, and ease of maintenance, accounting for a significant proportion of pipeline systems. Buried pipelines frequently traverse a variety of extraction zones, posing significant threats to pipeline safety. In these mining zones, pore collapse subjects pipelines to complex mechanical loads, including tensile, compressive, shear, torsional, and bending stresses. When stress concentration causes materials’ strength limits to be exceeded, pipeline rupture may ultimately occur. To enhance operational safety, pipelines crossing mining areas must implement online safety monitoring. This monitoring provides critical early warning for large-scale deformation and rupture risks triggered by ground subsidence and settlement.

Modern long-distance buried pipeline monitoring systems primarily incorporate three key approaches: leakage detection, peripheral vibration monitoring, and structural deformation monitoring. (1) Leakage monitoring: Pipeline leaks induce localized changes in soil temperature and thermal conductivity. Temperature monitoring systems detect these thermal anomalies to precisely locate leakage points [1,2,3]. (2) Vibration monitoring: Distributed fiberoptic vibration sensors deployed along pipelines continuously collect vibration signals from the surrounding environment. A vibration sample space is pre-established through the acquisition of various actual vibration signals and deep learning methods. By comparing samples within this space, the system identifies adverse vibration loads such as excavator digging, drilling operations, and encroachment, enabling timely early warnings [4,5,6,7]. (3) Deformation monitoring: The most critical safety monitoring technology utilizes stress–strain sensors to measure pipeline deformation. This approach enables early warning of potential leakage disasters caused by excessive deformation. Given the flammable and explosive nature of pipeline contents, traditional electrical sensors present safety risks. Optical fiber sensing solutions, particularly fiber Bragg grating (FBG) and distributed optical fiber sensor, offer significant advantages: intrinsically safe passive operation and long-term monitoring capability. Among these sensing technologies, FBG sensors provide localized high-accuracy measurements at critical stress points [8,9,10,11,12]. For example, Wu et al. designed a FBG strain sensor with a non-intrusive structure which can greatly enhance the strain sensing coefficient [13]. Cabral et al. proposed a robust and cost-effective approach to monitor pipeline bonded joints during assembly and operation using fiber Bragg grating sensors embedded into the joints’ adhesive layer [14]. The extensive operational mileage of pipelines makes missed detections common when monitoring pipeline damage using fiberoptic grating sensors. Distributed fiber sensors enable comprehensive coverage for continuous strain measurement. For example, He et al. used a distributed optical fiber Brillouin sensor to measure the strain of a pipeline and reconstructed pipe morphology based on the conjugate beam principle [15]. Lalam et al. proposed a natural gas pipeline monitoring method based on deep neural networks and distributed optical fiber sensors [16]. Compared to fiber Bragg grating sensing technology, distributed fiber Brillouin strain sensing systems exhibit lower strain measurement accuracy and less flexible deployment methods. To integrate the advantages of fiber Bragg gratings’ high local precision with distributed fiber’s large-area strain measurement capabilities, researchers have proposed co-linear fiber Bragg grating and distributed fiber technology. This approach enables both detailed measurements at critical structural locations and high-precision monitoring across extensive structural areas [17]. In the existing pipeline monitoring process, the monitoring parameters are related to vibration, temperature and strain, etc. In existing pipeline monitoring systems, key parameters primarily involve vibration, temperature, and strain. Relying solely on a single parameter to assess pipeline operational safety makes it difficult to accurately gauge the pipeline’s true condition and increases the likelihood of false damage alerts. For instance, pipeline monitoring technologies based on vibration data face significant challenges with regard to accurately identifying vibration patterns, resulting in high false alarm rates in practical engineering applications. Additionally, pipelines crossing mined-out areas face more complex safety challenges. The expansion of these areas causes redistribution of soil stress and soil vibration. By monitoring and identifying soil vibrations during the expansion process, early warnings of pipeline deformation can be achieved. Pipeline deformation monitoring provides a more intuitive reflection of pipeline safety status. Furthermore, simultaneous measurement of pipeline vibration and deformation enables timely warnings of adverse vibrations above the pipeline, thereby avoiding false vibration alarms.

To enhance pipeline safety in mining areas, this study proposes an integrated safety monitoring method based on vibration–strain fusion analysis. Firstly, we designed a pipeline deformation test to carry out research on pipeline deformation monitoring methods with an uncertain force direction based on fiberoptic grating sensors; then, we constructed a model test of pipelines throughout the mining zone to carry out research on pipeline safety monitoring methods during the process of expanding the mining zone based on fiber optic vibration and deformation information.

## 2. Principle of the Fiberoptic Strain and Vibration Sensing Technologies

Considering both localized fine monitoring and large-scale continuous monitoring of the pipeline, in our test, we deployed fiber grating in the most unfavorable position in the pipeline and distributed fiberoptics in the length direction of the pipeline. Our aim was to obtain information about the pipeline, such as the local strain, continuous strain and vibration. To this end, below, we briefly introduce the sensing principles behind fiber gratings, distributed Brillouin strain fiber, and distributed vibration fiber.

### 2.1. Principle of FBG Sensing Technology

Fiber Bragg grating is a passive device, the central wavelength of which is linear with strain and temperature, and the linear relationship can be expressed as follows:(1)Δλ=Δε+ΔT
where Δλ is the central wavelength variation of FBG; Δε,ΔT are the strain and temperature variations, respectively; and $C_{\varepsilon }(=1.2\;pm/\mu \varepsilon )$ and $C_{T}(C_{T}=10.8\;pm/{}^{\circ}C)$ are the strain and temperature sensitivity coefficients, respectively [18,19].

The center wavelength of the FBG is sensitive to both temperature and strain, so an FBG temperature sensor is needed to compensate for the effects of ambient temperature on the strain sensor. The FBG temperature sensor only senses temperature, which is expressed as follows:(2)$${\Delta \lambda }_{T}=C_{TT}\Delta T$$

Based on Equations (1) and (2), we can calculate the strain as follows:(3)$$\Delta \varepsilon =(\Delta \lambda -\frac{{\Delta \lambda }_{T}C_{T}}{C_{TT}})/C_{\varepsilon }$$

### 2.2. Principle of Distributed Brillouin Optical Fiber Sensing Technology

Based on analysis of the Brillouin scattering spectrum, it can be determined that the Brillouin frequency shift is linear with both the strain and the temperature, which can be expressed as follows:(4)$$\Delta v_{B}=C_{\varepsilon }\Delta \varepsilon +C_{T}\Delta T$$
where $C_{\varepsilon },C_{T}$ denote the strain and temperature sensitivity coefficients, respectively. We have conducted Brillouin frequency shift–strain and Brillouin frequency shift–temperature calibration tests in the lab, and the strain and temperature sensitivity coefficients are 0.05 MHz/με and 1.00 MHz/°C, respectively [20,21].

We also know that the optical fiber Brillouin frequency shift is sensitive to both temperature and strain, so we need the optical fiber temperature sensor to be close to the strain sensor for ambient temperature compensation. The optical fiber temperature sensor only senses temperature, which is expressed as follows:(5)$$\Delta v_{BT}=C_{TT}\Delta T$$

Based on Equations (4) and (5), the strain can be calculated as follow:(6)$$\Delta \varepsilon =(\Delta v_{B}-\frac{\Delta v_{BT}C_{T}}{C_{TT}})/C_{\varepsilon }$$

In this paper, all the tests were conducted indoors. The ambient temperature remained essentially constant, so we did not compensate for the effects of temperature on the strain measurements.

### 2.3. Principle of the Distributed Optical Fiber Vibration Sensing Technology

In the following test, phase-sensitive optical time domain reflectometry (Φ-OTDR) was used to measure the vibration signals. The pressure applied to the vibration optical fiber sensor caused a change in the elastic light effect which made the length and the refractive index of the optical fiber change, and then caused a phase change in the transmitted light in the optical fiber.

When the optical fiber is free of external force, we have(7)$$E_{out}=E_{in}\exp[j(\frac{2\pi nL}{c})]$$

Here, $E_{out},E_{in}$ are the outgoing and incident waves, respectively; n,L,c are the refractive index, the length of the optical fiber, and the light velocity, respectively.

Suppose the additional phase induced by external stress is Δφ(t), then the $E_{out}$ can be expressed as follows:(8)$$E_{out}=E_{in}\exp[j(\frac{2\pi nL}{\lambda +\Delta \varphi (t)})]$$(9)$$\Delta \varphi (t)=\beta L\frac{\Delta L}{L}+L\frac{\partial \beta }{\partial n}\Delta n+L\frac{\partial \beta }{\partial \alpha }\Delta \alpha$$

Here, λ is the wavelength of light; β is the propagation constant of light waves in optical fibers; and α is the radius of the optical fiber [22,23].

## 3. Pipeline Deformation Monitoring Test Under Unknown Direction

### 3.1. Sensor Layout

The buried pipeline belongs to the underground hidden project, and its force direction has a certain level of uncertainty. In order to correctly determine the deformation of the pipeline, four FBG strings named FBG1, FBG2, FBG3, and FBG4, respectively, were installed on the surface of the PVC pipe, as shown in Figure 1. Each FBG string included five FBG sensors, and the naming rules for FBGs in FBG1 are FBG11, FBG12, FBG13, FBG14 and FBG15. FBG1, FBG2, and FBG3 were placed at 90° intervals along the axial direction of the PVC pipe, and FBG2 and FBG4 were spaced at 30° intervals. Here, the spacing between adjacent FBGs is 500 mm, and the FBGs (produced by Dalian Bo Ruixin Technology Co., Ltd., Dalian, China) are bonded to the PVC pipe using epoxy resin adhesive.

Theoretically, the strain value at the symmetrical position of PVC pipe is equal in size and opposite in direction in the elastic range. It is assumed that the measured strain values of FBG1, FBG2, and FBG3 under a given load are ε_1_, ε_2_ and ε_3_, and the strain error due to the current ambient temperature change is ε_t_. After removing the environmental effects, we can obtain the true strain values at each location:ε_1z_ = −ε_2z_ = (ε_1_ − ε_2_)/2(10)ε_t_ = (ε_1_ + ε_2_)/2(11)ε_3z_ = ε_3_ − (ε_1_ + ε_2_)/2(12)

Here, ε_1z_, ε_2z,_ and ε_3z_ are the true strain values of FBG1, FBG2, and FBG3.

Assuming that the angle between the direction of force on the pipe and the horizontal direction is θ, we have(13)$$\theta =arctan\left(\frac{\varepsilon_{1z}}{\varepsilon_{3z}}\right)$$

And the maximum strain ε_max_ on the pipe is the combined strain of ε_1z_ and ε_3z_ expressed as the following equation:(14)$$\varepsilon_{max}=\sqrt{\varepsilon_{1z}^{2}+\varepsilon_{3z}^{2}}$$

Figure 2 shows a photograph of the sensor deployment at the middle position of the PVC pipe. The initial center wavelengths of each of the four FBG strings are 1530.4256 nm, 1535.1376 nm, 1545.0032 nm, 1550.2413 nm, and 1555.7829 nm, respectively.

### 3.2. Deformation Measurement Test of the PVC Pipe

Figure 3 is the schematic diagram and test photo of PVC pipe model. The diameter and length of the PVC pipe are 63 mm and 4000 mm, respectively. The PVC pipe is solidly supported at both ends, and the loading method involves applying graded loads by suspending weights at the midspan of the PVC pipe, and each load level is 500 g, with a total of three load levels.

The initial state of the PVC pipe is that FBG2 is located above the PVC pipe. In the next test, we designed two test conditions by turning the PVC pipe: in the first condition, FBG4 was located on the upper surface of the PVC pipe, i.e., the force direction of the PVC pipe was perpendicular to FBG4; for the second condition, the force direction of the PVC pipe was at an angle of 45° with FBG1, as shown in Figure 4. Three levels of loading were applied to the PVC pipe for each condition.

In loading condition 1, the PVC pipe is rotated at an angle of 30° with respect to the initial state of the PVC pipe, so the FBG4 measures the maximum compressive strain of the PVC pipe. Based on Equations (11)–(14), the maximum strain (Max-strain) and the rotation angle can be calculated. In this condition, under each load level, the FBG4 measurement indicates maximum negative strain, while the value calculated from Equation (14) indicates maximum positive strain (Max-strain). Theoretically, the absolute values of these two should be equal. Figure 5 shows the test results of loading condition 1. As can be seen from Figure 5a–d, the absolute value of the strain measured by FBG4 is essentially consistent with the maximum strain value, with a maximum error below 5.2%, and the angular errors at each measurement point are within 4.2% at all levels of loading, as shown in Figure 5e. In loading condition 2, because the load direction and FBG2 and FBG3 are all at an angle of 45°, and FBG2 and FBG3 are located in the tensile zone of the pipeline, theoretically, the strain values measured by FBG2 and FBG3 are equal in size with positive strain, and the strain value measured by FBG1 is a negative strain. Figure 6 shows the test results for loading condition 2. As can be seen from the figure, the angular errors at each measurement point are within 2.8% at all levels of loading. Based on the above test results, it can be determined that the direction of PVC pipe deformation and the maximum strain value can be calculated from the measured values of three FBG strings placed at 90° intervals.

## 4. Pipeline Vibration and Deformation Monitoring Test Through the Mining Area

### 4.1. Test Introduction

Figure 7 shows the schematic diagram of the pipeline model in the mining area. The pipeline is PVC material, with a diameter, length, and elastic modulus of 50 mm, 9000 mm and 1.25 GPa. Fixed supports are set every 3 m on the pipeline, and a movable support is set up between the two fixed supports to simulate the expansion of the soil hole in the mining area. Figure 8 shows the test photo, which includes a DVS (Distributed Vibration Sensing) system, a BOTDR (Brillouin Optic Time Domain Reflectometer)) system, and an FBG system. Here, the measurement accuracies of the BOTDR and FBG systems are ±20 με and ±2 με, and the positioning accuracies of the BOTDR and DVS systems are 500 mm and 1500 mm, respectively.

In the test, a mass block is applied to the PVC pipeline, and the vibration response is generated by the shaking of the mass block and the movement of the support; at the same time, the mass block acts on the PVC pipeline, and the PVC pipeline is deformed as shown in Figure 9. As shown in Figure 9, each mass is a rectangular steel block, with an approximate mass of 2000 g, and it is suspended from a PVC pipe by a wire. In order to obtain the vibration and strain information, the optical fiber is pasted on the upper and lower surfaces of the pipeline, while at the same time, a 5 m free optical fiber is set at the fixed support position of the pipeline. Three FBG strain sensors were mounted at the mid-span position of the PVC pipeline, midway between the two fixed supports. Here, the grating section length is 15 mm, and they are bonded to the PVC pipe using epoxy resin adhesive.

### 4.2. Experimental Process and Data Analysis

Figure 10 shows the test procedure. Initially, the movable support was located 1 m from the fixed support, and a mass block was suspended at 25 cm, 50 cm, and 75 cm from the fixed support; then, each time the movable support was moved to the right by 25 cm to simulate the expansion of the soil hole. At the same time, a mass block was added at the location of the previous movable support. Eventually, the movable support was 2 m away from the fixed support.

Figure 11a,c present the vibration information during the movement of the movable support and Figure 11d shows the vibration signals generated when adding a mass block. It can be seen that the OF can monitor the vibration signals applied on the PVC pipeline. After the DVS system tests the vibration signal, the vibration fiber is connected to the BOTDR system as a distributed strain fiber to test the current strain value of the PVC pipe, and the strain distribution is shown as in Figure 12. By comparing Figure 11 and Figure 12, it can be seen that the location of the maximum strain and the location of the maximum vibration signal of the PVC pipe are basically the same.

Figure 13 shows a schematic diagram of simultaneous loading of three zones on a PVC pipe. Table 1 shows the center wavelength values of the three FBGs and the corresponding strain values on the PVC pipe. Figure 14 shows the strain values measured by FBG sensors and OF sensors (Marked with a red line). Since FBG-S1 and FBG-S2 are located near the movable support, their measured strains are not the maximum strain in the section, while FBG-S3 is located exactly at the maximum deformation of the PVC pipe in its current state, so its measured value is the largest.

In actual engineering applications, buried pipelines undergo seasonal temperature fluctuations and exposure to precipitation and seepage during their service, resulting in significant environmental temperature variations throughout their operational lifespan. For bare optical gratings, a 1 °C change in ambient temperature can introduce an error of approximately 10 με. Since the thermal expansion coefficient of steel is greater than that of optical fiber, the resulting strain measurement error is amplified. Therefore, temperature sensors must be deployed near the pipeline to provide temperature compensation for the strain sensors. Furthermore, in actual engineering practice, we implement a two-tiered warning strategy: If vibration signals are detected but no pipeline deformation signals are recorded, this indicates that the goaf is expanding, while the overlying soil layer exhibits minimal deformation insufficient to cause pipeline displacement. This triggers a Level 1 warning. When both vibration signals and pipeline deformation signals are detected simultaneously, it signifies that the soil cavity has expanded to a significant extent, causing substantial deformation in the overlying soil layer that leads to pipeline displacement. This triggers a Level 2 warning, requiring intervention to address both the soil cavity expansion and pipeline deformation.

## 5. Conclusions

In this paper, we proposed a safety monitoring method for pipelines through the mining area by integrating optical fiber strain sensing and distributed optical fiber vibration sensing technologies. The method of monitoring the main stress direction and maximum stress of a PVC pipe by laying three FBG strings at 90° intervals on the PVC pipe was studied. We also constructed a pipeline model through the mining area, and carried out the monitoring method of vibration response and pipeline deformation during the extension process of the mining area based on a local FBG strain sensor, BOTDR strain sensor, and DVS sensor. The following results were obtained:The FBG strings laid at 90° intervals can correctly identify the force direction of the PVC pipe and obtain the maximum strain value of the PVC pipe, and the error of the test with two different angles of rotation is within 5.2%.Results from the goaf expansion simulation test indicate that distributed fiberoptic vibration sensors can effectively identify and locate vibration signals during the expansion process of soil cavities. Meanwhile, distributed fiberoptic strain sensors and localized fiber Bragg grating sensors can accurately measure the deformation of PVC pipelines. Based on the vibration and strain data collected during goaf expansion, a two-tiered early warning strategy for pipeline damage has been established.

In actual engineering projects, although the initial expansion of soil cavities may cause redistribution of stress in the roof soil, the resulting soil deformation may be minimal, making it difficult for strain sensors installed within the pipeline to detect such minute deformations. At this stage, vibration signals from soil expansion can provide early warnings for pipeline safety. Simultaneously, as the soil cavity continues to expand, significant soil deformation occurs, causing pipeline deformation. This enables precise assessment of pipeline deformation through strain measurement.

## Figures and Tables

**Figure 1 micromachines-16-01074-f001:**
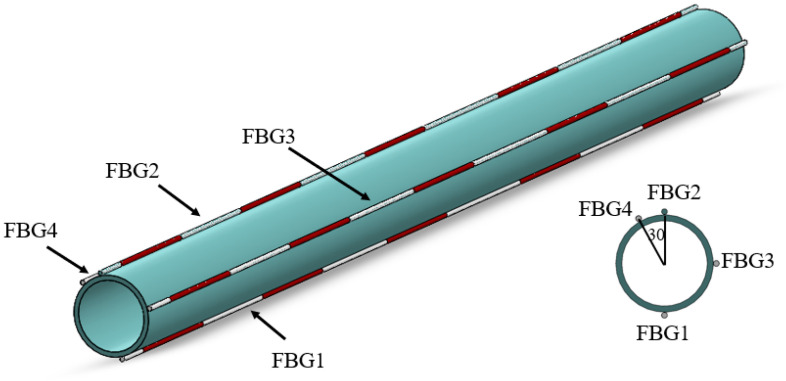
Sensor layout on the PVC pipe.

**Figure 2 micromachines-16-01074-f002:**
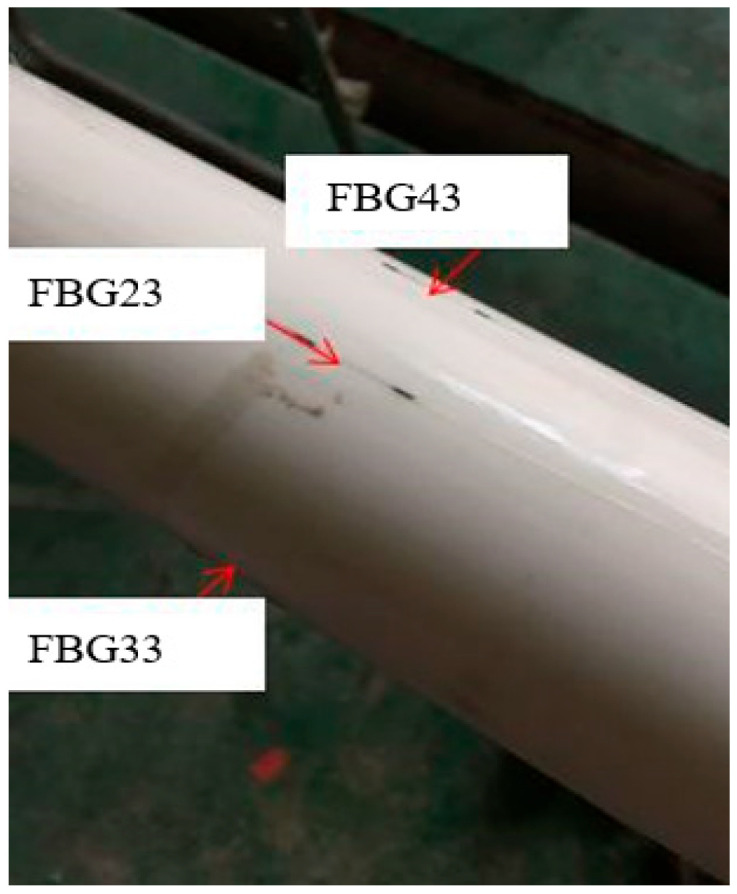
The photograph of the sensor deployment.

**Figure 3 micromachines-16-01074-f003:**
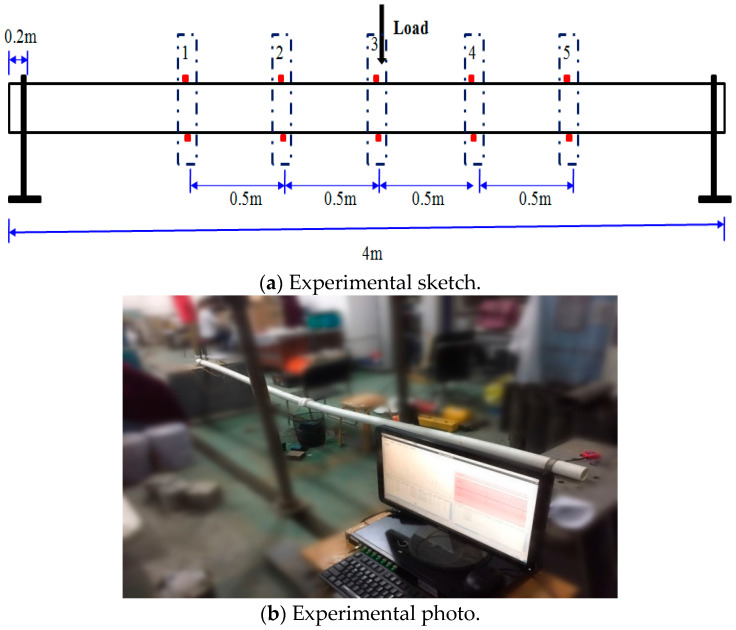
Setup of the deformation measurement test.

**Figure 4 micromachines-16-01074-f004:**
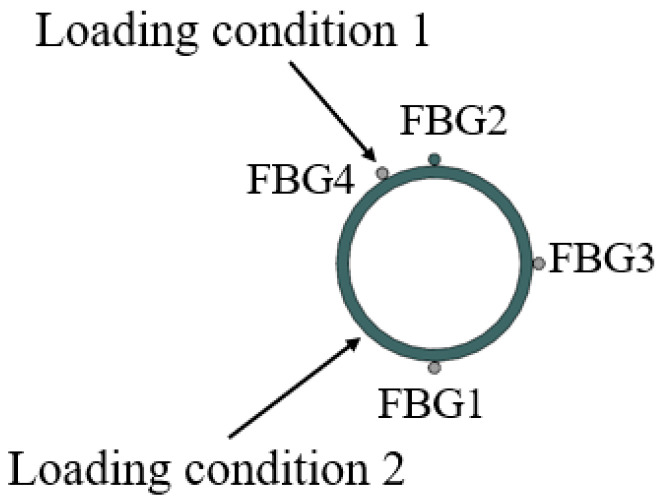
Loading condition of the PVC pipe test.

**Figure 5 micromachines-16-01074-f005:**
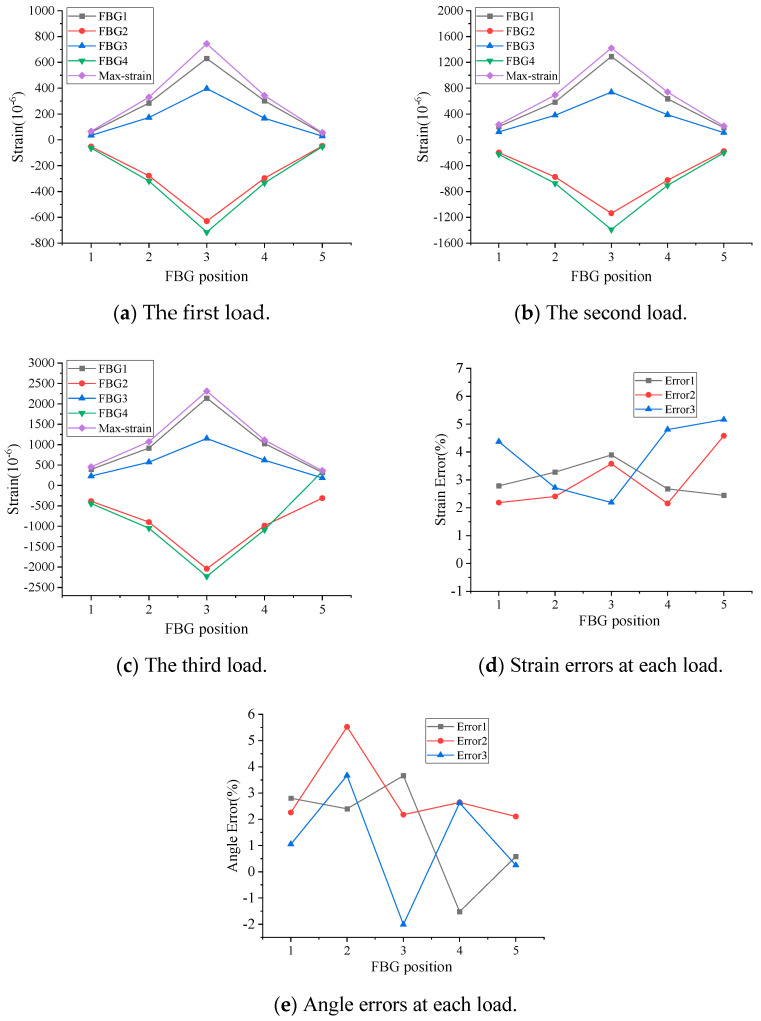
Test results for loading condition 1.

**Figure 6 micromachines-16-01074-f006:**
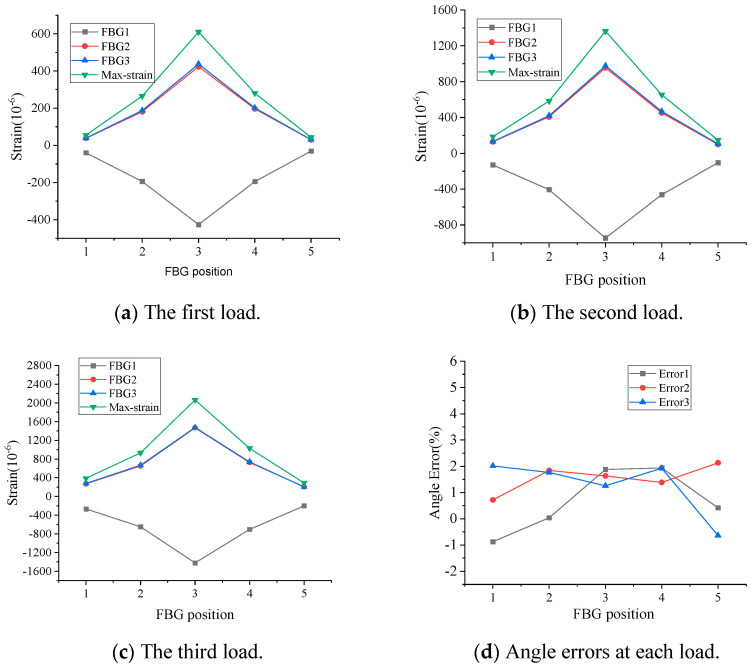
Test results for loading condition 2.

**Figure 7 micromachines-16-01074-f007:**
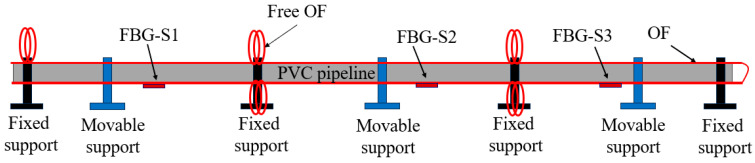
Schematic diagram of the pipeline model in the mining area.

**Figure 8 micromachines-16-01074-f008:**
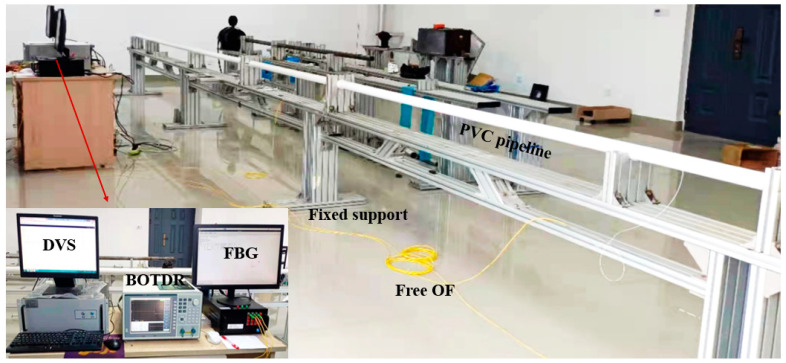
A photo of the experimental setup.

**Figure 9 micromachines-16-01074-f009:**
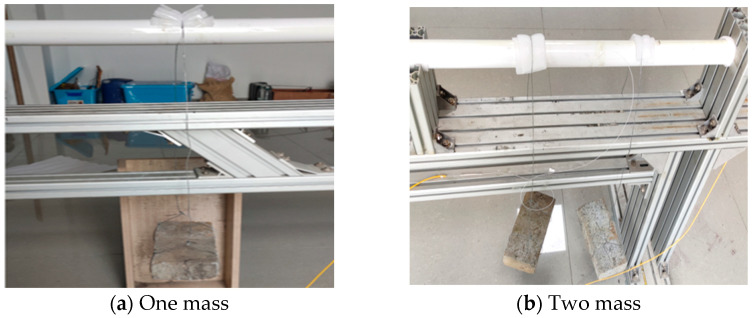
Photo of the mass block loading.

**Figure 10 micromachines-16-01074-f010:**
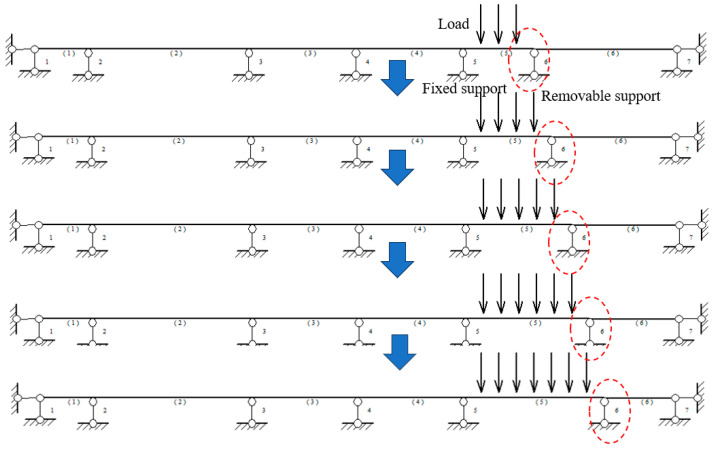
Test procedure.

**Figure 11 micromachines-16-01074-f011:**
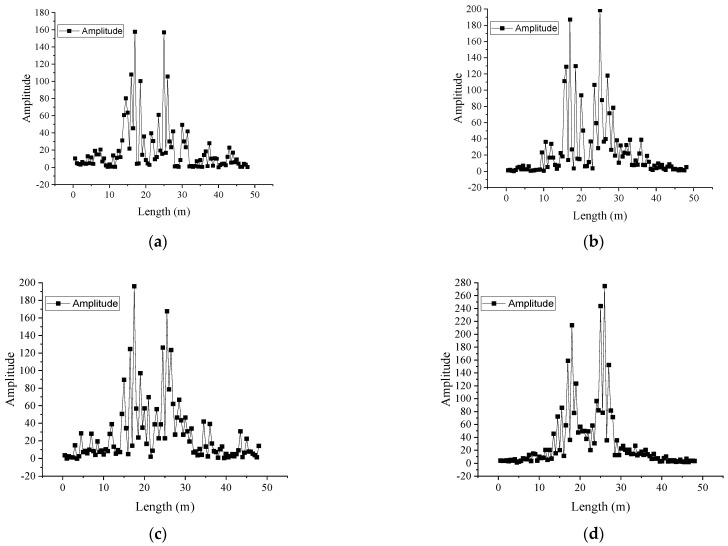
Vibration signals during loading and movement of the support. (**a**) Vibration signals generated by moving the movable support from 125 cm to 150 cm. (**b**) Vibration signals generated by moving the movable support from 150 cm to 175 cm. (**c**) Vibration signals generated by moving the movable support from 175 cm to 200 cm. (**d**) Vibration signals generated when adding a mass block.

**Figure 12 micromachines-16-01074-f012:**
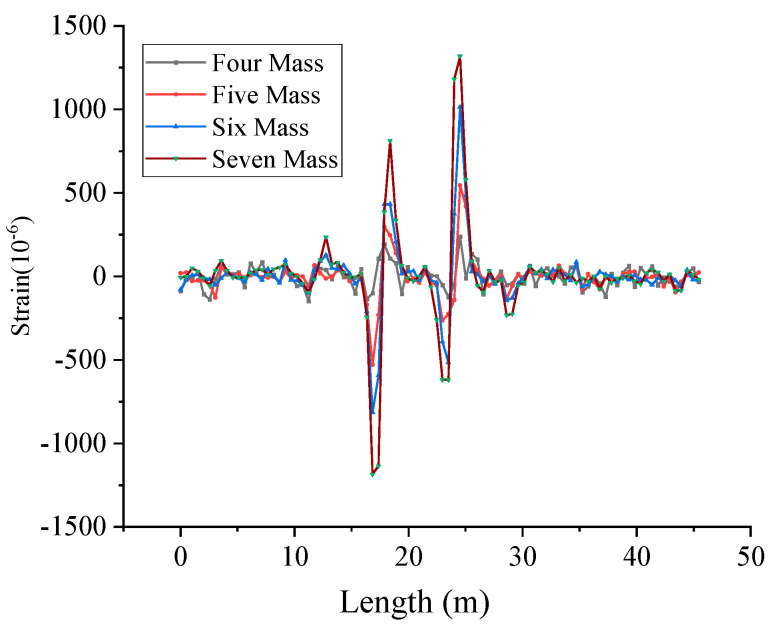
Strain distribution measured by BOTDR system.

**Figure 13 micromachines-16-01074-f013:**
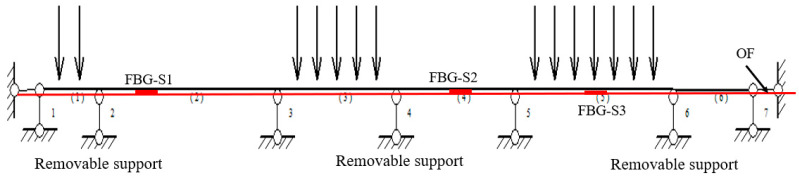
Schematic diagram of simultaneous loading of three zones.

**Figure 14 micromachines-16-01074-f014:**
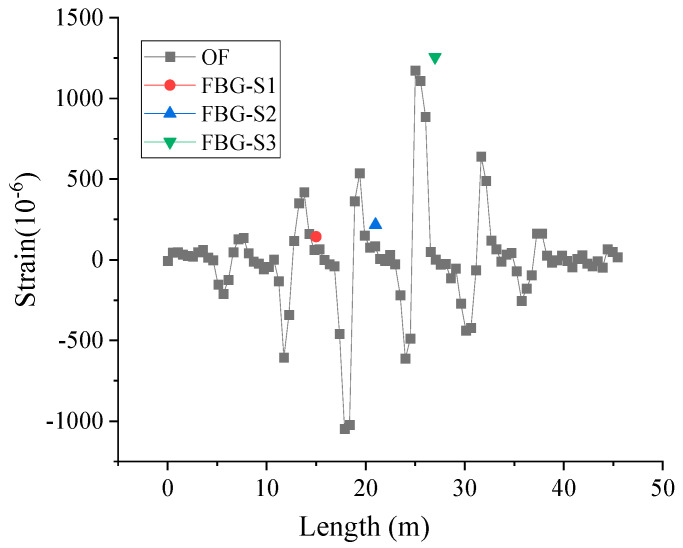
Strain information measured by FBG and OF sensors.

**Table 1 micromachines-16-01074-t001:** Information about the FBG sensor.

	FBG-S1	FBG-S2	FBG-S3
Initial center wavelength (nm)	1539.9610	1535.1750	1545.4680
Final center wavelength (nm)	1540.1324	1535.4352	1546.9745
Strain increment (10^−6^)	142.83	216.83	1255.42

## Data Availability

The original contributions presented in the study are included in the article; further inquiries can be directed to the corresponding author.

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
