# Peer review of "Safety Monitoring Method for Pipeline Crossing the Mining Area Based on Vibration–Strain Fusion Analysis"

_micromachines, 2025, doi:10.3390/mi16091074_

Round 1

Reviewer 1 Report

Comments and Suggestions for Authors

This paper presents a combined fiber-optic strain and distributed vibration monitoring method for pipeline safety assessment in mining areas and demonstrates strong engineering application potential. I have carefully reviewed the manuscript and recommend that the authors consider the following points, with the hope that these suggestions will be helpful in improving the paper.

  1. In real engineering scenarios, pipelines often experience multi-point loading with varying force directions along their length. It is unclear how the proposed 90° FBG configuration can accurately resolve local force directions and magnitudes under such conditions, and whether additional sensor arrangements or computational models are necessary.

  1. In practical applications, vibration, strain, and temperature changes typically occur simultaneously. A detailed explanation of the data fusion or signal separation techniques used to separate these effects and ensure measurement accuracy is necessary.

  1. This study proposes early-warning capability based on vibration monitoring; however, the criteria for setting alarm thresholds are unclear. The authors should clarify how these thresholds are determined for pipelines with different materials, operating conditions, and burial environments, and whether field validation has been conducted.

  1. The 9 m PVC pipeline model differs from actual mining-area pipelines in terms of material, burial depth, and soil–structure interaction. The potential influence of these differences on monitoring accuracy and the measures taken to extrapolate model results to real engineering conditions should be discussed.

  1. The formatting of the references is inconsistent: some journal names are italicized while others are not, and some author names appear to be misspelled. It is recommended that the authors carefully review and correct these issues.

  1. Mathematical symbols should follow standard typographical conventions: functions (e.g., cos, exp), and constants (e.g., π) must be typeset in upright font.It should also be noted that the sizes of the equations in the manuscript are inconsistent and need to be standardized.
Comments on the Quality of English Language

The overall readability of the manuscript is good; however, some sentences contain grammatical errors. It is recommended to perform a thorough grammar check.

Author Response

This paper presents a combined fiber-optic strain and distributed vibration monitoring method for pipeline safety assessment in mining areas and demonstrates strong engineering application potential. I have carefully reviewed the manuscript and recommend that the authors consider the following points, with the hope that these suggestions will be helpful in improving the paper.

Comments 1:

In real engineering scenarios, pipelines often experience multi-point loading with varying force directions along their length. It is unclear how the proposed 90° FBG configuration can accurately resolve local force directions and magnitudes under such conditions, and whether additional sensor arrangements or computational models are necessary.

Response 1: Thanks. In practical applications, due to the unique operational characteristics of buried pipelines, sensors are typically installed on the upper surface of the pipeline. The effectiveness of this installation scheme has only been verified through PVC pipe deformation tests. The specific verification method involves applying deformation in a specific direction, then identifying it using sensors spaced at 90-degree intervals. The identification results are compared with sensors installed in that direction, yielding an identification error below 6%. Results are shown in Figures 5 and 6 in the revised paper. Of course, in subsequent research, we will validate the effectiveness of this method on a newly constructed pipeline.

Comments 2:

In practical applications, vibration, strain, and temperature changes typically occur simultaneously. A detailed explanation of the data fusion or signal separation techniques used to separate these effects and ensure measurement accuracy is necessary.

Response 2: Thanks, this is a highly specialized question.

On the one hand, FBG strain sensors are typically affected by ambient temperature, necessitating environmental temperature compensation for pipeline strain measurements. We presented fiber Bragg grating strain compensation methods in Chapter 2. All experiments discussed herein were conducted indoors under essentially constant ambient temperatures, thus environmental temperature compensation was not considered. On the other hand, adverse vibrations near pipelines often indicate potential pipeline damage. However, it is difficult to quantify pipeline damage using vibration signals alone. Identifying adverse vibration signals near pipelines is a complex issue. Simultaneous measurement of strain and vibration signals can significantly enhance pipeline early warning efficiency. In the revised paper, we have explained this in the section of “Introduction” as follows:

“While current monitoring systems track vibration, temperature, and strain parameters to assess pipeline safety, mining zones present unique challenges: increased hazards compared to normal geological conditions; mining expansion causes soil vibration and stress redistribution; and single-parameter monitoring proves inadequate for comprehensive safety assessment.”

Comments 3:

This study proposes early-warning capability based on vibration monitoring; however, the criteria for setting alarm thresholds are unclear. The authors should clarify how these thresholds are determined for pipelines with different materials, operating conditions, and burial environments, and whether field validation has been conducted.

Response 3: Thanks, this is an engineering problem. This paper proposes an early warning method based on vibration-strain information. Due to variations in pipeline materials, installation conditions, and operating environments, it is not feasible to provide a definitive threshold value. The research findings presented herein have not yet been applied to actual pipelines. In subsequent projects, we will carefully consider the reviewers' suggestions and conduct threshold analysis and validation for pipeline early warning systems tailored to specific operational pipelines.

 Comments 4:

The 9 m PVC pipeline model differs from actual mining-area pipelines in terms of material, burial depth, and soil–structure interaction. The potential influence of these differences on monitoring accuracy and the measures taken to extrapolate model results to real engineering conditions should be discussed.

Response 4: Thanks, that's a great question. For model testing, a 9-meter-long pipeline represents an exceptionally long pipe model. The model differs significantly from actual pipeline installation conditions, as it does not account for burial depth, soil-structure interaction, or the propagation characteristics of vibration signals within the soil. The primary objective of this paper is to propose a vibration-deformation monitoring method that utilizes monitoring data to reflect damage in pipeline models traversing mined-out areas.

 Comments 5:

The formatting of the references is inconsistent: some journal names are italicized while others are not, and some author names appear to be misspelled. It is recommended that the authors carefully review and correct these issues.

Response 5: Thanks, we have corrected these mistakes in the revised paper.

 Comments 6:

Mathematical symbols should follow standard typographical conventions: functions (e.g., cos, exp), and constants (e.g., π) must be typeset in upright font. It should also be noted that the sizes of the equations in the manuscript are inconsistent and need to be standardized.

Response 6: Thanks, we have corrected these mistakes in the revised paper.

Overall, the reviewer is a highly professional scholar who raised numerous constructive questions regarding the content of this paper. They demonstrated significant interest in the practical feasibility of the research for engineering applications. The methodology presented in this paper will be applied in our subsequent pipeline monitoring projects, and the findings will be further organized and published at that time.

Reviewer 2 Report

Comments and Suggestions for Authors

This paper studied the civil engineering applicability of a fiber sensing system composing of multiple FBGs, BOTDR and φ-OTDR systems for the health monitoring of PVC pipe line. The work was done experimentally only under constant temperature condition but provided some interesting experimental results which can promise future technical expansion of the fiber-optic pipe line heath monitoring system. For field use of your sensing system, short comments are desirable on surrounding temperature effect and on the effect of pipe-line’s longitudinal deformation including its actual occurrence.

Comments on the Quality of English Language

This paper studied the civil engineering applicability of a fiber sensing system composing of multiple FBGs, BOTDR and φ-OTDR systems for the health monitoring of PVC pipe line. The work was done experimentally only under constant temperature condition but provided some interesting experimental results which can promise future technical expansion of the fiber-optic pipe line heath monitoring system. For field use of the authors' sensing system, short comments are desirable on surrounding temperature effect and on the effect of pipe-line’s longitudinal deformation including its actual occurrence.

Author Response

Comments :

This paper studied the civil engineering applicability of a fiber sensing system composing of multiple FBGs, BOTDR and φ-OTDR systems for the health monitoring of PVC pipe line. The work was done experimentally only under constant temperature condition but provided some interesting experimental results which can promise future technical expansion of the fiber-optic pipe line heath monitoring system. For field use of your sensing system, short comments are desirable on surrounding temperature effect and on the effect of pipe-line’s longitudinal deformation including its actual occurrence.

Response : Thanks. at the end of the fourth section of the revised paper, we have elaborated on this issue as follows:

In actual engineering applications, buried pipelines undergo seasonal temperature fluctuations and exposure to precipitation and seepage during service, resulting in significant environmental temperature variations throughout their operational lifespan. For bare optical gratings, a 1°C change in ambient temperature can introduce an error of approximately 10 με. Since the thermal expansion coefficient of steel is greater than that of optical fiber, the resulting strain measurement error is amplified. Therefore, temperature sensors must be deployed near the pipeline to provide temperature compensation for the strain sensors.

Reviewer 3 Report

Comments and Suggestions for Authors

The manuscript “Safety monitoring method for pipeline crossing the mining area based on vibration-strain fusion analysis” used three different fiber optical sensors including (1) Fiber Bragg grating (FBG), (2) distributed Brillouin optical fiber sensing technology and (3) distributed optical fiber vibration sensing technology to monitor the stress and vibration of a PVC pipeline by simulating the conditions of mining areas. The paper did a good experimental design, and the results from the three methods are interrelated to each other. I would suggest its publication on Micromachines with the following revisions:

  1. In the introduction part, please explain why 3 types of sensors should work together for monitoring the PVC pipeline, and what is the benefit of using all of them at the same time.
  2. The sentence in lines 77-78, “To this end, the following are respectively on the fiber grating, distributed Brillouin strain fiber and distributed vibration fiber sensing principle is briefly introduced” can be more concisely revised to " To this end, the following briefly introduces the sensing principles behind fiber gratings, distributed Brillouin strain fiber, and distributed vibration fiber".
  3. In lines 106 and 107, “𝛥𝑣𝐵𝑇” should be “𝛥𝜆𝑇”.
  4. In section 3.1, please explain where the FBGs are purchased from and what the pitch is for the FBGs, what is the type of fiber for the FBG and how it is fabricated if the FBGs are self-developed ones. How are the FBGs attached to the PCV pipe?
  5. In line 136, it is stated that “FBG2 and FBG4 are spaced at 60° intervals”. In Figure 1, it shows the interval is 30°. Please clarify. Also in line 136, “intervals” should be “interval”.
  6. In line 160, please explain how the force is loaded. What is the quantified amount of force loaded, and what is the exact location that the force is loaded?
  7. Please indicate how the angular error and strain error at each measurement points are identified in Figures 5(d) and 6(d). The strain error is not shown in Figures 5(d) and 6(d), please add them in Figures 5(d) and 6(d).
  8. Why FBG4 is not measured in Figure 6?
  9. In Figure 5, the caption for (a) should be “The first load”, not “Table 5”.
  10. In lines 190-191, please introduce the FBG system, the BOTDR system and DVS system, including the instruments used, light source and light power, etc.
  11. In line 194, what does each mass block look like (size and weight, material), how are they applied to the PVC pipe?
  12. In line 198-199, how long is the optical fiber with FBG strain sensors? How are they mounted on the PVC pipeline (e.g., by what type of paste)?
  13. Please label FBGs S1-S3 (which are introduced in lines 221-224) inside Figure 13.
  14. To understand Figure 14 well, please label the position of OF in Figure 13.

Author Response

The manuscript “Safety monitoring method for pipeline crossing the mining area based on vibration-strain fusion analysis” used three different fiber optical sensors including (1) Fiber Bragg grating (FBG), (2) distributed Brillouin optical fiber sensing technology and (3) distributed optical fiber vibration sensing technology to monitor the stress and vibration of a PVC pipeline by simulating the conditions of mining areas. The paper did a good experimental design, and the results from the three methods are interrelated to each other. I would suggest its publication on Micromachines with the following revisions:

Response: We sincerely appreciate the reviewers' recognition of the article's content. We will carefully revise the manuscript based on their suggestions.

Comments1:

In the introduction part, please explain why 3 types of sensors should work together for monitoring the PVC pipeline, and what is the benefit of using all of them at the same time.

Response1: In the introduction section, we have added the following contents to explain the issue.

“In existing pipeline monitoring systems, key parameters primarily involve vibration, temperature, and strain. Relying solely on a single parameter to assess pipeline operational safety makes it difficult to accurately gauge the pipeline's true condition and increases the likelihood of false damage alerts. For instance, pipeline monitoring technologies based on vibration data face significant challenges in accurately identifying vibration patterns, resulting in high false alarm rates in practical engineering applications. Additionally, pipelines crossing mined-out areas face more complex safety challenges. The expansion of these areas causes redistribution of soil stress and soil vibration. By monitoring and identifying soil vibrations during the expansion process, early warnings of pipeline deformation can be achieved. Pipeline deformation monitoring provides a more intuitive reflection of pipeline safety status. Furthermore, simultaneous measurement of pipeline vibration and deformation enables timely warnings of adverse vibrations above the pipeline, thereby avoiding false vibration alarms.”

Comments2:

The sentence in lines 77-78, “To this end, the following are respectively on the fiber grating, distributed Brillouin strain fiber and distributed vibration fiber sensing principle is briefly introduced” can be more concisely revised to " To this end, the following briefly introduces the sensing principles behind fiber gratings, distributed Brillouin strain fiber, and distributed vibration fiber".

Response2: Thanks for the good suggestion. In the revised paper, we have corrected it.

Comments3:

In lines 106 and 107, “????” should be “???”.

Response3: Thanks. In lines 106 and 107,“????” is correct, because this section describes the temperature compensation method for distributed fiber Brillouin sensor, not the temperature compensation method for fiber Bragg gratings.

Comments4:

In section 3.1, please explain where the FBGs are purchased from and what the pitch is for the FBGs, what is the type of fiber for the FBG and how it is fabricated if the FBGs are self-developed ones. How are the FBGs attached to the PCV pipe?

.

Response4: Thanks. we have added the following content in the section 3 in the revised paper.

Here, the spacing between adjacent FBGs is 500mm, and the FBGs (produced by Dalian Bo Ruixin Technology Co., Ltd., ) are bonded to the PVC pipe using epoxy resin adhesive.

Comments5:

In line 136, it is stated that “FBG2 and FBG4 are spaced at 60° intervals”. In Figure 1, it shows the interval is 30°. Please clarify. Also in line 136, “intervals” should be “interval”.

Response5: Thanks. It’s our mistakes. The interval between the FBG2 and FBG4 should be 30°. We have corrected it in the revised paper.

Comments6:

In line 160, please explain how the force is loaded. What is the quantified amount of force loaded, and what is the exact location that the force is loaded?

Response6: Thanks. we have added the loading information in the revised paper as follows.

“the loading method involves applying graded loads by suspending weights at the midspan of the PVC pipe, and each load level is 500g, with a total of three load levels.”

Comments7:

Please indicate how the angular error and strain error at each measurement points are identified in Figures 5(d) and 6(d). The strain error is not shown in Figures 5(d) and 6(d), please add them in Figures 5(d) and 6(d).

Response7: Thanks.

In the figure 5. We have added a strain error diagram (as shown in figure5d), where the strain error is obtained by comparing the FBG4 measurement values with the maximum calculated error, and the maximum strain error is beyond 5.2%.

Figure 6 shows the test results of condition 2. In this condition, the strain values measured by the FBG4 are no longer the maximum negative strain, so we do not provide the maximum strain error here; only the angular error is given. We have removed the erroneous description of strain error.

Comments8:

Why FBG4 is not measured in Figure 6?

Response8: Under these operating conditions, the strain value measured by FBG4 is no longer the maximum negative strain. Since the maximum value and angle of the PVC pipe can be derived solely from the other three FBG measurement values, the measurement from FBG4 is not plotted in Figure 6.

Comments9:

In Figure 5, the caption for (a) should be “The first load”, not “Table 5”.

Response9:

Thanks, we have corrected the mistakes.

Comments10:

In lines 190-191, please introduce the FBG system, the BOTDR system and DVS system, including the instruments used, light source and light power, etc.

Response10:

Thanks, we have added the sensing performances of the BOTDR, DVS and FBG system as follows in the revised paper.

Here, the measurement accuracy of the BOTDR and FBG system are ±20με and±2με, and the positioning accuracy of the BOTDR and DVS system are 500mm and 1500mm respectively.

Since we are using commercial equipment, the specific light source and light power of the equipment are not particularly relevant to our experiment, so they have not been listed.

Comments11:

In line 194, what does each mass block look like (size and weight, material), how are they applied to the PVC pipe?

Response11: As shown in figure 9, each mass is approximately a rectangular steel block, with a mass of 2000g and it is suspended from a PVC pipe by a wire. We have added the information in the revised paper.

Comments12:

In line 198-199, how long is the optical fiber with FBG strain sensors? How are they mounted on the PVC pipeline (e.g., by what type of paste)?

Response12:

The grating section length is 15mm, and they are bonded to the PVC pipe using epoxy resin adhesive.

We have added the information in the revised paper.

Comments13:

Please label FBGs S1-S3 (which are introduced in lines 221-224) inside Figure 13.

Response13:

The grating section length is 15mm, and they are bonded to the PVC pipe using epoxy resin adhesive.

We have added the information in the revised paper.

Comments14:

Please label FBGs S1-S3 (which are introduced in lines 221-224) inside Figure 13.

Response14:

We have marked the sensors on the Figure 13.

Comments15:

To understand Figure 14 well, please label the position of OF in Figure 13.

Response15:

We have marked the sensors on the Figure 13.

Reviewer 4 Report

Comments and Suggestions for Authors

Comments:

  • The authors state in their introduction: "Vibration monitoring: Distributed fiber-optic vibration sensors installed along the pipeline continuously acquire vibration signals from the surrounding environment." However, the article lacks explanation of how to use fiber-optic sensors for dynamic load monitoring. Authors are encouraged to refer to articles by Dr. Ying's research group, for example: https://www.sciencedirect.com/science/article/abs/pii/S0030399224000112.
  • In Figure 1, several FBGs are connected at different locations. How did the authors choose these locations? Are these locations critical along the pipeline?
  • There is a typographical error in Equation (14).
  • The authors state that " The initial center wavelengths of each of the four FBG strings are 1530nm, 1535nm, 1545nm, 1550nm and 1555nm respectively." These values ​​should be the theoretical Bragg wavelength lengths. What is the actual initial wavelength of each FBG sensor?
  • In Figure 4, the FBGs are not deployed symmetrically. What is the engineering significance of each FBG position in the loading test? What do they simulate?
  • Section 4.2, “Experimental Process and Data Analysis,” lacks further analysis related to actual operating conditions. When operators are able to obtain signals from various sensors, what information can they obtain from the signals? What measures should be taken for maintenance or replacement?

Author Response

Comments1:

The authors state in their introduction: "Vibration monitoring: Distributed fiber-optic vibration sensors installed along the pipeline continuously acquire vibration signals from the surrounding environment." However, the article lacks explanation of how to use fiber-optic sensors for dynamic load monitoring. Authors are encouraged to refer to articles by Dr. Ying's research group, for example: https://www.sciencedirect.com/science/article/abs/pii/S0030399224000112.

Response1:

Thanks. In the introduction section, we have added the vibration load monitoring method, at the same time, we cited the Dr. ying’s paper in the revised paper.

Vibration Monitoring: Distributed fiber optic vibration sensors deployed along pipelines continuously collect vibration signals from the surrounding environment. A vibration sample space is pre-established through the acquisition of various actual vibration signals and deep learning methods. By comparing samples within this space, the system identifies adverse vibration loads such as excavator digging, drilling operations, and encroachment, enabling timely early warnings [4-7]

Comments2:

In Figure 1, several FBGs are connected at different locations. How did the authors choose these locations? Are these locations critical along the pipeline?

Response2:

Thanks. This is an excellent question. In practice, the closer the spacing between fiber optic gratings, the higher the measurement accuracy, but the cost also increases. In this experiment, we deployed fiber optic gratings at 500mm intervals, primarily because PVC pipe material is relatively uniform. Within its elastic range, the strain between two grating points exhibits a largely linear relationship. In actual service conditions, the fiber optic grating should be positioned at the most unfavorable location. For this test, the most unfavorable position was determined to be the midspan location.

Comments3:

There is a typographical error in Equation (14).?

Response3:

Thanks. we have corrected it.

Comments4:

The authors state that " The initial center wavelengths of each of the four FBG strings are 1530nm, 1535nm, 1545nm, 1550nm and 1555nm respectively." These values should be the theoretical Bragg wavelength lengths. What is the actual initial wavelength of each FBG sensor?

Response4: In the original text, we only provided the integer part of the initial center wavelength of the fiber Bragg grating. Following the reviewer's suggestion, we have supplemented the fractional part of the fiber Bragg grating.

Comments5:

In Figure 4, the FBGs are not deployed symmetrically. What is the engineering significance of each FBG position in the loading test? What do they simulate?

Response5: In the diagram, FBG1, FBG2, and FBG3 are arranged at 90° intervals. These three fiber Bragg gratings primarily identify the direction and magnitude of pipeline deformation in different orientations. FBG4 serves primarily as a reference point. When the force direction acts on FBG4, the strain value for that force direction is calculated using FBG1, FBG2, and FBG3. This calculated strain value is then compared with the measurement from FBG4 to validate the effectiveness of this method.

Comments6:

Section 4.2, “Experimental Process and Data Analysis,” lacks further analysis related to actual operating conditions. When operators are able to obtain signals from various sensors, what information can they obtain from the signals? What measures should be taken for maintenance or replacement?

Response6:

Thanks. this is a practical engineering application problem. In the revised paper, we have established the following early warning mechanism.

In actual engineering practice, we implement a two-tiered warning strategy: If vibration signals are detected but no pipeline deformation signals are recorded, this indicates the goaf is expanding while the overlying soil layer exhibits minimal deformation insufficient to cause pipeline displacement. This triggers a Level 1 warning. When both vibration signals and pipeline deformation signals are detected simultaneously, it signifies the soil cavity has expanded to a significant extent, causing substantial deformation in the overlying soil layer that leads to pipeline displacement. This triggers a Level 2 warning, requiring intervention to address both the soil cavity expansion and pipeline deformation.

Round 2

Reviewer 1 Report

Comments and Suggestions for Authors

The article has been appropriately revised. What I mentioned earlier is also fully complemented. Thank you to the authors again for answering the questions and performing the revision requested. I think the paper will be ready for publication in Micromachines.

Reviewer 4 Report

Comments and Suggestions for Authors

Accept.